# Extracellular Vesicles for Childhood Cancer Liquid Biopsy

**DOI:** 10.3390/cancers16091681

**Published:** 2024-04-26

**Authors:** Nilubon Singhto, Pongpak Pongphitcha, Natini Jinawath, Suradej Hongeng, Somchai Chutipongtanate

**Affiliations:** 1Ramathibodi Comprehensive Cancer Center, Faculty of Medicine Ramathibodi Hospital, Mahidol University, Bangkok 10400, Thailand; nilubon.sin@mahidol.ac.th; 2Bangkok Child Health Center, Bangkok Hospital Headquarters, Bangkok 10130, Thailand; pongpak.po@bangkokhospital.com; 3Division of Hematology and Oncology, Department of Pediatrics, Faculty of Medicine Ramathibodi Hospital, Mahidol University, Bangkok 10400, Thailand; suradej.hon@mahidol.ac.th; 4Program in Translational Medicine, Faculty of Medicine Ramathibodi Hospital, Mahidol University, Bangkok 10400, Thailand; natini.jin@mahidol.ac.th; 5Chakri Naruebodindra Medical Institute, Faculty of Medicine Ramathibodi Hospital, Mahidol University, Samut Prakan 10540, Thailand; 6Integrative Computational Biosciences Center, Mahidol University, Nakon Pathom 73170, Thailand; 7MILCH and Novel Therapeutics Laboratory, Division of Epidemiology, Department of Environmental and Public Health Sciences, University of Cincinnati College of Medicine, Cincinnati, OH 45267, USA; 8Extracellular Vesicle Working Group, University of Cincinnati College of Medicine, Cincinnati, OH 45267, USA

**Keywords:** brain tumor, childhood cancer, diagnosis, extracellular vesicles, exosomes, liquid biopsy, molecular cargo, precision medicine, prognosis, treatment monitoring

## Abstract

**Simple Summary:**

Liquid biopsy is a technique that uses minimally invasive or noninvasive methods to detect disease biomarkers in biofluids, such as blood and urine. For childhood cancers, this approach is promising, especially when tissue biopsies are challenging. By analyzing tiny particles called extracellular vesicles, the small particles in biofluids which carry genetic and molecular information from cancer cells, this liquid biopsy technique can aid early diagnosis, treatment monitoring, and outcome prediction, potentially improving the outcomes of childhood cancers. This review article introduces the concept of extracellular vesicle liquid biopsy, summarizes the progress that is being made in the diagnosis of multiple types of pediatric malignancies, and provides prospects for using this extracellular vesicle method to guide the development of novel cancer therapeutics.

**Abstract:**

Liquid biopsy involves the utilization of minimally invasive or noninvasive techniques to detect biomarkers in biofluids for disease diagnosis, monitoring, or guiding treatments. This approach is promising for the early diagnosis of childhood cancer, especially for brain tumors, where tissue biopsies are more challenging and cause late detection. Extracellular vesicles offer several characteristics that make them ideal resources for childhood cancer liquid biopsy. Extracellular vesicles are nanosized particles, primarily secreted by all cell types into body fluids such as blood and urine, and contain molecular cargos, i.e., lipids, proteins, and nucleic acids of original cells. Notably, the lipid bilayer-enclosed structure of extracellular vesicles protects their cargos from enzymatic degradation in the extracellular milieu. Proteins and nucleic acids of extracellular vesicles represent genetic alterations and molecular profiles of childhood cancer, thus serving as promising resources for precision medicine in cancer diagnosis, treatment monitoring, and prognosis prediction. This review evaluates the recent progress of extracellular vesicles as a liquid biopsy platform for various types of childhood cancer, discusses the mechanistic roles of molecular cargos in carcinogenesis and metastasis, and provides perspectives on extracellular vesicle-guided therapeutic intervention. Extracellular vesicle-based liquid biopsy for childhood cancer may ultimately contribute to improving patient outcomes.

## 1. Introduction

The evolution of pathological measures from histomorphology and immunohistochemistry to molecular diagnostics has dramatically enhanced the accuracy of cancer diagnosis and classification. This advancement offers more precise prognostication and tailors therapeutic plans, including targeted therapy and immunotherapy, for individual cancer patients [1]. To obtain tumor tissues for diagnosis, invasive procedures are required, including surgical removal or biopsy. These procedures come with risks of anesthesia complications, bleeding, and infection [2]. Furthermore, the inherent heterogeneity of tumor cells and their clonal evolution over time often necessitate repeated biopsies from primary, metastatic, or relapsed tumors, adding additional risk to patients [3]. Tissue biopsies may also introduce sampling bias, leading to an incomplete analysis of tumor subpopulations. Moreover, tumors in some areas, such as the brain stem, are inaccessible for biopsy due to the unacceptable risk of mortality [4].

Liquid biopsy has emerged as a minimally invasive approach supporting diagnosis, making therapeutic decisions, and monitoring various types of cancers when surgical or biopsy procedures are unavailable or unacceptable [5]. This approach also holds potential for early screening and monitoring of minimal residual disease [6]. Liquid biopsy captures tumor fragments or circulating tumor contents from body fluids such as blood [7], cerebrospinal fluid [8], ascites [9], saliva [10], bronchoalveolar lavage [11], and urine [12]. This allows for repeated assessments during diagnosis, treatment, and follow-up periods with less invasiveness and minimized complications [13,14]. Therefore, liquid biopsy allows the possibility of detecting overall subclonal tumor heterogeneity throughout the clinical course [15].

Blood and plasma are reliable sources for most clinical investigations, and the same is true for cancer liquid biopsy [16]. For example, peripheral blood liquid biopsy-next-generation sequencing tests have been approved by the Food and Drug Administration (FDA) since 2020 to identify *BRCA1/2, ALK*, *PIK3CA*, and *ATM* pathogenic variants as part of the diagnosis of breast, ovarian, and lung cancers and treatment guidance (https://www.fda.gov/) [17]. Studies have demonstrated the potential applicability of blood-based liquid biopsies in childhood cancers [18], i.e., neuroblastoma, osteosarcoma, Ewing sarcoma, and hematologic malignancies [18]. Urine collection is generally less uncomfortable than taking blood and does not require trained medical staff. Urine liquid biopsies have shown promise in tracking somatic mutations in pediatric patients with Wilms tumors and other childhood renal tumors for diagnosis, risk stratification, and monitoring treatment response [19]. The blood–brain barrier blocks the transmigration of tumor tissue fragments, e.g., cell-free DNA (cfDNA), between systemic circulation and the central nervous system (CNS), thus making CSF a viable option for developing liquid biopsies for primary brain tumors and CNS metastases [20]. Various techniques and platforms have been developed for CSF liquid biopsy, for instance, detecting histone H3 mutation [21] and cfDNA [22,23] in CSF for diagnosis and monitoring of gliomas and medulloblastoma [20]. Ascites and pleural effusion can be good resources for liquid biopsies to determine treatment responses in advanced lung and gastrointestinal malignancies [24], while saliva is an invaluable source for head and neck cancers [25,26].

Cancer liquid biopsy platforms mainly rely on cell-related extracellular components, including circulating tumor cells (CTCs), circulating tumor DNA (ctDNA), and tumor-derived extracellular vesicles (EVs), each of which has unique advantages and challenges (Table 1).

CTCs are cancer cells that detach from the primary tumor and invade systemic circulation [27]. CTCs can appear as individual cells or clusters, with the latter being associated with a higher metastasis potential [28]. Various methods have been developed to detect CTCs [29], including immunocytochemical methods [30], fluorescence-activated cell sorting (FACS) [31], reverse transcription-quantitative polymerase chain reaction [32], fluorescence in situ hybridization (FISH) [33], and mass spectrometry [27]. Nonetheless, the main challenge of CTC liquid biopsy is based on the scarce number of CTCs in circulation, i.e., patients with advanced metastasis show up to 10 CTCs per 1 mL of peripheral blood [28]. In addition, CTC detection provides insights into cancer-related invasiveness and metastasis rather than information on genetic mutations, translocations, or rearrangements [27,34].

CtDNA is typically shorter and more fragmented than non-tumor DNA, representing 0.01–10% of total cell-free DNA in the bloodstream [34] and carrying various cancer-specific molecular markers, such as single-nucleotide mutations, methylation, and cancer-derived viral sequences [35]. A unique advantage of ctDNA is that it requires a smaller biofluid sample for assessing tumor mutational burden. Studies indicated that 1 mL of plasma is sufficient for ctDNA monitoring for tumor response to treatment [36,37], making it particularly useful for pediatric patients, where obtaining large sample volumes can be difficult. CtDNA-based liquid biopsy has been investigated in various types of pediatric solid tumors, e.g., neuroblastoma [38], Ewing sarcoma [39], osteosarcoma [40], medulloblastoma [14], renal tumors [41], rhabdomyosarcoma [42], and retinoblastoma [43]. However, ctDNA has a short half-life of two hours since it is degraded by extracellular DNase enzymes and rapidly cleared from systemic circulation by various organs, including the kidneys and liver [44]. The low abundance of cancer-specific ctDNA and the corresponding high abundance of background DNA in circulation also pose a considerable challenge in early tumor stage detection [45].

Recent studies increasingly point to EVs as a critical player in liquid biopsy diagnostics [46,47,48]. EVs have gained attention due to their unique characteristics of carrying various molecular cargos, i.e., lipids, proteins, and nucleic acids of origin cells [46,49]. Thus, EVs play a crucial role in cell-to-cell communication mediated through ligand–receptor interaction and membrane fusion to release molecular contents, e.g., growth factors, kinases, enzymes, and non-coding RNAs, all of which can elicit changes and reprogram intracellular activities of the recipient cells [50]. Moreover, EVs can reflect the altered physiological and pathological states of their parent cells and organs. This fact opens new possibilities for circulating EVs in cancer liquid biopsies [46,49]. Note that EVs presented in biofluids are heterogeneous, consisting of multiple subpopulations released from various cell sources overlapping particle size and density. Thus, a challenge in employing EVs for liquid biopsy lies in the complexities of EV isolation and characterization, necessitating meticulous optimization at the initial phase of any project. However, when compared with CTCs and ctDNA, EVs offer several advantages for liquid biopsies. First, EVs are abundant in biofluids, approximately 1 × 10^9^ particles/mL, making it relatively easier to collect sufficient quantities for analysis [51]. Second, EVs are actively secreted by living cells, including tumor cells, and carry a diverse range of information reflective of ongoing activities within their parent cells [51]. In contrast, ctDNA primarily releases from apoptotic or dead tumor cells and may provide less accurate information [51]. Finally, the lipid bilayer-enclosed EV structure provides protection to molecular cargos from proteases, DNases, and RNases in biofluids, allowing them to circulate in the bloodstream from 30 min to 6 h without degradation [49,52].

In this review, according to the promising characteristics of EVs for liquid biopsy, we summarize the EV definition and essential methods for isolation and characterization and then extensively explored the research involving the potential of EVs as biomarkers for diagnosis, treatment monitoring, and prognostic prediction in pediatric malignancies. Additionally, we discusse the mechanistic roles of EVs in carcinogenesis and metastasis, and provide our perspectives on future directions of EV-based liquid biopsy for childhood cancer.

**Table 1 cancers-16-01681-t001:** Tissue fragments detectable by cancer liquid biopsy.

Tissue Fragment	Description	Analytical Techniques	Advantages	Limitations	References
Circulating tumor cells (CTCs)	Cancer cells detach from the primary tumor or metastatic lesion and circulate in the bloodstream.	Fluorescence-activated cell sorting (FACS) Reverse transcription-quantitative polymerase chain reaction (RT-qPCR) Fluorescence in situ hybridization (FISH) Mass spectrometry	CTCs contain DNA, RNA, and proteins. CTC molecular profiling can reveal clonal evolution, heterogeneity, invasiveness, and metastasis. CTCs provide insights into cancer-related phenomena.	The scarcity of CTCs in the blood is sometimes as rare as one per 10^5^–10^7^ mononuclear cells. Limitation of their diagnostic sensitivity, especially in non-metastatic cases.	[27,28,29,31,32,33]
Circulating tumor DNA (ctDNA)	ctDNA is shed by primary tumor cells and subsequently enters the bloodstream. As such, ctDNA carries the genetic alterations present in the original tumor.	Droplet digital polymerase chain reaction (ddPCR) Beads, emulsion, amplification, and magnetics (BEAMing) PCR Tagged-amplicon deep sequencing (TAm-Seq) Reverse transcription-quantitative polymerase chain reaction (RT-qPCR) Next-generation sequencing (NGS)	ctDNA requires a smaller biofluid sample for assessing tumor mutational burden. The presence of genomic tumor alterations is determined in ctDNA. The methylation patterns of cell-free nucleic acid are consistent with their respective cells or tissues. Cell-free non-coding RNAs (cf-ncRNAs) are very stable because they are associated with proteins or enclosed in vesicles.	The sensitivity to detect mutations is inadequate in cases where the mutant allele fraction is low. The low abundance of cancer-specific ctDNA and the corresponding high abundance of background DNA in circulation poses a considerable challenge in early tumor stage detection. Cell-free nucleic acid may be released from dead cells, not living cells.	[35,36,37,41,45,53,54,55]
Extracellular vesicles (EVs)	Tiny membranous structures enclosed by a lipid bilayer are secreted by both normal and cancerous cells. EVs contain a diverse assortment of molecular constituents, including DNA, RNA, non-coding RNA, and proteins.	Microscopy (e.g., electron microscopy (EM) and atomic force microscopy (AFM)) Nanoparticle tracking analysis (NTA) Dynamic light scattering (DLS) Enzyme-linked immunosorbent assay (ELISA) Western blot analysis Mass spectrometry NGS	EVs are plentiful in biofluids (1 × 10^9^ particles/mL), making it easy to collect enough for further analysis. EVs contain a wide variety of information that accurately represents their originating cells. The lipid bilayer structure makes EVs stable in biofluids, allowing them to circulate in the hostile tumor microenvironment.	High degree of variability in the methodologies employed for isolation procedures. Absence of standardized isolation protocols. Easy contamination of non-EV particles or soluble proteins and difficulty of EV characterization.	[51,56,57,58]

## 2. EV Definition and Methods: A Brief Summary

The International Society for Extracellular Vesicles (ISEV) has recently launched minimal information for studies of extracellular vesicles 2023 (MISEV2023). In these new guidelines, the definition of EVs has been updated as “*the particles that are released from cells, are delimited by a lipid bilayer, and cannot replicate on their own*” [58]. Since it is likely that a broad population of EVs is being studied rather than a solitary subtype, biogenesis-related terms, including exosomes [59,60] and ectosomes [61,62], should not be used unless their subcellular origins can be demonstrated in their studies [58]. Ambiguous terms, i.e., exosome-like vesicles and microvesicles, are discouraged to avoid any nomenclature confusion in the field [58]. To address EV subtypes, operational terms are recommended based on the diameter of the separated particles: small EVs (sEVs; <200 nm) and large EVs (LEVs; >200 nm), where cautions should be made since the measured diameter is related to the specific characterization method [58]. In this review, we follow the MISEV2023 guideline and only use one umbrella term, EVs, when referring to any types of vesicles investigated in the original studies.

For EV isolation, the single-step procedure, i.e., high-speed ultracentrifugation (UC), polymer/precipitation agents (e.g., polyethylene glycol), and ultrafiltration (UF), offers high EV recovery from biofluids [63,64]. However, these techniques may lead to reduced specificity due to the contaminations of non-vesicular components (e.g., soluble proteins and lipoprotein micelles). In contrast, size-exclusion chromatography (SEC), asymmetrical flow field-flow fractionation, and immunoaffinity bead capture offer better specificity while ensuring reasonable recovery yields [63]. Nonetheless, when superior EV specificity is paramount, especially in studies focused on specific EV functions or subpopulations, a multi-step combinatorial method is recommended [65], for example, two or three-step combinations between a form of centrifugation, polymer precipitation, UF, and SEC [64,65,66,67]. These multi-step protocols ensure minimal contamination, yielding samples optimal for direct downstream analysis and data interpretation. However, their scalability and reproducibility must be addressed in the early phase of the project to ensure sufficient recovery yields with the high specificity of EVs for further characterizations.

After EV isolation, various techniques are used to characterize the purity of EVs, commonly based on morphology, hydrodynamic size, and protein markers [68]. Transmission electron microscopy (TEM) provides a high-resolution image of the nano-size EVs morphology [56]. However, morphology imaging may be affected by swelling, shrinking, or flattening from sample preparation steps, e.g., fixation, dehydration, embedding, and imaging conditions [56]. To overcome the above limitation, atomic force microscopy (AFM) and cryo-electron microscopy provide detailed information and solve the variable effects of TEM, but these techniques tend to be relatively low-throughput and resource-intensive [56]. In addition, EVs can be determined by depending on biophysical approaches. Several techniques, including nanoparticle tracking analysis (NTA), resistive pulse techniques, and dynamic light scattering, are used for determining the distribution of EV size [56]. However, these techniques have been reported to be sensitive to the size of protein or glycan extending from the EV membrane, and they do not provide the phenotype of vesicles. Protein evidence of EV presence in the isolates is usually demonstrated by three positive EV markers and one negative marker, including (i) surface proteins and CD9, CD63, and CD81 tetraspanins; (ii) proteins involved in EV biogenesis (such as the Tsg101, Alix, and Rab families); (iii) cytosolic proteins (e.g., Hsp70 and Gapdh); (iv) absence of proteins representing intracellular organelles (i.e., calnexin, GM130, and histone).

For more information about EV definition, isolation, and characterization, please refer to the ISEV guidelines for EV studies [57,58].

## 3. EV-Based Liquid Biopsy for Childhood Cancers

In the context of cancer, EVs play a role in facilitating metastasis by promoting the formation of a pre-metastatic niche (PMN) [69], which is the microenvironment at a distant organ that disseminates tumor cells from the primary tumor colonized and grows into metastasis [69]. Tumor-derived EVs prepare the PMN by modulating extracellular matrix formation, immune suppression, angiogenesis, and facilitating the recruitment of other stromal cells to support tumor growth and survival [69]. These EV molecular cargos serve as invaluable resource to discover biomarker candidates for several types of childhood cancer, including neuroblastoma, medulloblastoma, hepatoblastoma, osteosarcoma, Ewing’s sarcoma, rhabdomyosarcoma, and lymphoma. In the following section, we provide current evidence of EV studies conducted over recent decades with a special emphasis on transcripts and protein candidates for the development of childhood cancer liquid biopsy (summarized in Figure 1 and Table 2).

### 3.1. Pediatric Neuroblastoma

Neuroblastoma is the most common extracranial pediatric solid tumor, which arises from neural crest cells [99]. The clinical spectrum of neuroblastoma can vary from localized disease and spontaneous regression to highly malignant and extensive metastasis disease [99]. Genetic aberrations include recurrent segmental chromosome aberration (losses of chromosome 1p, 3p, 4p, 11q and gains of 1q, 2p, 17q), amplification of the proto-oncogene *MYCN,* and epigenetic modifications [99]. Although there have been significant improvements in the prognosis of neuroblastoma patients, up to 40% of high-risk cases continue to pose a substantial challenge, with a considerable portion experiencing tumor relapse driven by minimal residue disease (MRD) [100].

Ongoing research fields put effort into improving the management of high-risk neuroblastoma through early detection of MRD by liquid biopsy [100]. Interestingly, cancer-specific modification DNA and RNA can be detected in EVs. Therefore, it is theoretically presumed that MRD could be detected through cancer EVs [101]. The extensive studies of molecular cargos of EVs derived from neuroblastoma in different disease states should be more elucidated.

For example, our group recently reported that transcripts of the *MYCN* gene, the strongest molecular risk of neuroblastoma, were detectable in LEVs isolated from patient-derived bone marrow plasma, where the presence and absence of MYCN-LEVs were associated with *MYCN*-amplification status and treatment–relapse states [46]. Haug et al. [70] reported that neuroblastoma cell lines with *MYCN* amplification release EVs containing oncogenic miRNAs, including 11 types of miRNAs (e.g., miR-16, 125b, 21, 23a, 24, 25, 27b, 218, 320a, 320b, and 92a) [70], where miR-92a was the highest miRNA expression [70]. This study implied that these EVs might contribute to the aggressive behaviors of *MYCN*-amplified neuroblastoma [70]. Similarly, Ma et al. [71] performed next-generation sequencing on miRNAs containing EVs isolated from human plasma. This study revealed that high numbers of miRNA were changed by either upregulated or downregulated in patients with neuroblastoma and ganglioneuroblastoma compared to healthy donors [71]. Among these altered miRNAs, two, including miR-199a-3p and miR-495-5p, overlap in neuroblastoma and ganglioneuroblastoma patients [71]. After validation by RT-qPCR, they discovered that only miR-199a-3p expression was significantly higher expressed in neuroblastoma patients [71]. Interestingly, miR-199-3p represents an independent risk factor since its expression in the plasma was more elevated in neuroblastoma patients with unfavorable histology than in those with a good prognosis [71].

Using proteomic analysis, Coletti et al. [72] compared EVs derived from neuroblastoma cell lines representing primary tumors and bone marrow metastasis. Overall, EV proteins of primary tumors are involved in neuronal development and function, suggesting a connection between the neuronal characteristics of neuroblastoma and EV cargo. In contrast, proteins exclusively present in EVs derived from neuroblastoma–bone marrow metastatic models, e.g., signal peptidase complex catalytic subunit SEC11, cell division cycle-associated protein 3, nuclear pore complex protein Nup107, calcium, and integrin-binding protein 1, are associated with cell survival, proliferation, and progression [72]. These proteins could be further developed as potential biomarkers of neuroblastoma tumor staging. Collectively, these studies suggested that EV-based liquid biopsy is useful for diagnosis, monitoring of treatment response/relapsed disease, and prognosis prediction in pediatric neuroblastoma [46,70,71,72].

### 3.2. Medulloblastoma

Medulloblastoma is the predominant malignant brain tumor in children, accounting for approximately 20% of all pediatric brain tumors. It is classified as an embryonal neuroepithelial tumor of the cerebellum [102]. According to the fifth edition of the WHO classification of tumor of the CNS, medulloblastoma incorporates both molecular subgroups and traditional variants to provide a more comprehensive understanding of the disease biology [103], including the wingless (WNT)-activated subgroup, sonic hedgehog (SHH)-activated subgroup, Group 3, and Group 4 [103]. In addition, the SHH-activated subgroup can be subdivided by the detection of *TP53* gene mutation, which includes SHH-activated, *TP53* wild-type, and SHH-activated *TP53* mutants [103]. Nonetheless, the heterogeneity of the clinical course and outcome in each patient poses challenges regarding the lack of early diagnostic biomarkers for high-risk patients with recurrent and metastatic disease.

Kaid et al. [73] performed proteomic and miRNA analysis and revealed that 464 proteins and 10 microRNAs were exclusively detected in LEVs released from highly aggressive medulloblastoma cell lines. The interactome analysis of distinct proteins and miRNA suggested that ERK, PI3K/AKT/mTOR, EGF/EFGR, and stem cell self-renewal are the main oncogenic signaling pathways altered in aggressive medulloblastoma [73]. Four proteins (UBE2M, HNRNPCL2, HNRNPCL3, and HNRNPCL4) and five miRNAs (miR-4449, 500b, 3648, 1291, and 3607) were identified as the candidate biomarkers of aggressive medulloblastoma with potential applications in diagnosis, patient stratification, and early detection of relapsed disease [73]. In this direction, Zhu et al. [74] reported that increased levels of EV-derived miR-181a-5p, miR-125b-5p, and let-7b-5p were associated with in vitro invasion and migratory abilities of SHH medulloblastoma cells through the activation of ERK in the Ras/MAPK pathway [74].

Furthermore, a recent study by Jackson et al. [75] demonstrated that metastatic medulloblastoma cells released a higher amount of EVs compared to non-metastatic cells, and metastatic EVs significantly increased surface matrix metalloproteinase-2 (MMP-2). Importantly, this study found a high level of MMP-2 activity in CSF from three of four patients associated with tumor progression [75], suggesting that MMP-2-expressed EVs may modulate the PMN to drive medulloblastoma metastasis [75].

### 3.3. Pediatric Gliomas

Gliomas are the most common histologic type of brain tumor, can arise in all areas of the brain hemisphere, and are rarely seen at the spinal cord level [104]. Regarding the WHO Classification for CNS neoplasm, a comprehensive molecularly refined diagnosis is crucial for therapeutic plans and prognoses [104]. However, tissue biopsies in some areas, such as deep hemispheric lesions, optic pathways, brain stem, and spinal cord, are limited due to unacceptable risk for morbidity and mortality [104]. In this regard, liquid biopsy can make a significant contribution to glioma molecular diagnosis despite brain tumors in difficult-to-access areas [104]. Molecular alteration targets, such as *H3*, *BRAF*, *EGFR*, and *IDH1* mutations, and other markers that could be used for diagnosis, monitoring treatment response, or developing novel therapeutics, have been investigated [104].

H3 mutations in diffuse high-grade gliomas are important for diagnosis and prognosis, while *BRAF* alterations in low-grade gliomas are valuable in both diagnosis and treatment. García-Romero et al. [105] demonstrated the feasibility of the detection of *BRAF* V600E targetable mutation by digital-PCR from cell-free-DNA and EV-derived DNA in serum, plasma, and CSF isolated from a cohort of 29 CNS pediatric patients. Chen et al. [76] used a novel approach that combines biofluid EV-RNA and BEAMing RT-PCR to detect and quantify mutant and wild-type *IDH1* RNA transcripts in the CSF of patients with gliomas.

Several studies suggested that EV-derived miRNA could be the biomarker for patient classification, predict response to treatment, or serve as a novel therapeutic target for pediatric gliomas. Xiao et al. [77] detected the EV-derived miR-301a from glioblastoma multiforme (GBM) cells under hypoxic conditions that promoted radiation resistance in GBM. MiR-301a is involved in regulating the Wnt/β-catenin pathway and targeting anti-oncogene TCEAL7. Therefore, these findings might facilitate the search for a novel target to overcome radiotherapy resistance in GBM patients. Ailiang et al. [78] demonstrated that a lower expression of miR-151a was related to temozolomide (TMZ) resistance in GBM; the restoration of miR-151a expression sensitized TMZ-resistant GBM cells by inhibiting XRCC4-mediated DNA repair. The detection of EV-derived miR-151a might predict chemotherapy response but also represents a promising therapeutic target for therapy-refractory GBMs. Additionally, Jianxing et al. [79] revealed that overexpression of miR-1238 led to the required resistance against TMZ in GBM patients, while another EV-derived miRNA, miR-148a, was associated with a risk of GBM progression according to The Cancer Genome Atlas [106].

### 3.4. Hepatoblastoma

Hepatoblastoma is a rare and aggressive type of liver cancer that primarily affects infants and children younger than three years old [107]. This malignancy is reported to arise from abnormal developments in the liver during embryonic stages [107]. The pathogenesis of hepatoblastoma involves genetic and molecular alterations that disrupt the normal processes of liver organogenesis, leading to the uncontrolled growth of cells and the formation of tumors [107]. Some genetic syndromes have been associated with an increased predisposition to hepatoblastoma, which includes Beckwith–Wiedemann syndrome, hemihypertrophy, and familial adenomatous polyposis [107]. The early diagnosis of hepatoblastoma in patients with a genetic predisposition to the disease is crucial for improving treatment outcomes, allowing for prolonged intervention and potentially more effective treatment strategies.

In this direction, Jiao et al. [80] conducted a transcriptomic study to identify EV-derived miRNAs as potential biomarkers for hepatoblastoma. As a result, EV-derived miR-34a/b/c was significantly lower in the serum of patients with hepatoblastoma compared to healthy control groups, where the downregulation of miR-34 was associated with tumorigenesis in the lungs, skin, breast, urinary bladder, and kidneys [108]. In hepatoblastoma, decreased levels of EV-derived miR-34a/b/c might be considered as the diagnostic and prognostic biomarker. Likewise, Liu et al. [81] revealed that the elevated level of miR-21 containing EV in hepatoblastoma patients might be another biomarker for hepatoblastoma, while Hu et al. [82] demonstrated EV-derived miR-126 was upregulated in hepatoblastoma cells. These findings suggest that EV-derived microRNA may be involved in liver cancer tumorigenesis.

### 3.5. Osteosarcoma

Osteosarcoma is the most common primary bone tumor affecting adolescents and young adults. It is characterized by a heterogenous malignant spindle cell tumor and the formation of immature osteoid tissue or osteoid and accounts for 1% of all cancers [109]. The current treatment approach for osteosarcoma involves a combination of neoadjuvant chemotherapy, surgical removal of the tumor, and adjuvant chemotherapy [110]. Despite intensive treatment regimens, the 5-year survival rate for patients with metastatic and recurrent osteosarcoma is less than 20 percent [111]. Therefore, osteosarcoma research is mainly focused on identifying new therapeutic targets to improve outcomes and early detection of metastasis.

Jarez et al. [87] investigated EV-miRNA expression in osteosarcoma cell lines with varying degrees of metastatic potential. The results showed four miRNAs (miR-21-5p, 143-3p, 148a-3p, and 181a-5p) were highly expressed in EVs derived from metastatic osteosarcoma cells compared to non-metastatic cells [87]. Gene ontology analysis of predicted miRNA target genes suggested that EVs may regulate the metastatic potential of osteosarcoma by inhibiting several genes (e.g., MAPK1, NRAS, FRS2, PRCKE, BCL2, and QKI) involved in apoptosis and cell adhesion [87]. In this direction, Araki et al. [86] revealed that sEVs derived from malignant human osteosarcomas contained miR-146a-5p and could inhibit osteoblastogenesis. Ucci et al. [112] revealed that osteosarcoma EVs had increased pro-osteoclastogenic proteins, inflammatory cytokines, and metalloproteinase, with decreased proteins involved in the cell cycle and pro-osteoblastogenesis. Moreover, Zhong et al. [89] reported that osteosarcoma with the RAB22a-NeoF1 fusion gene produced EVs containing the Rab22a-NeoF1 protein to promote PMN formation and M2 macrophage induction, eventually enhancing lung metastases [89]. Furthermore, Endo-Munoz et al. [90] demonstrated that elevation of the urokinase plasminogen activator (uPA) and the uPA receptor (uPAR) in both soluble form and osteosarcoma-secreted EVs were exclusive in metastatic osteosarcoma cells, where the uPA inhibitor significantly mitigated osteosarcoma metastasis in an orthotopic mouse model [90]. These findings suggested that osteosarcoma EVs contain molecular cargos with abilities to modulate the PMN, which contributes to osteosarcoma progression [86,89,90,112], thus serving as the basis for developing osteosarcoma liquid biopsy in future EV studies.

### 3.6. Ewing’s Sarcoma

Ewing’s sarcoma is the second most common primary bone malignancy affecting children and young adults between 10 and 20 years old. It typically arises in long bones and the pelvis and sees around 200 new cases annually in the United States [113]. The underlying driven pathogeneses are characterized by specific rearrangements of one of five alternative ETS family member genes, i.e., *FLI1* (90%), *ERG* (5–10%), *ETV1* (<1%), *E1AF* (<1%), and *FEV1* (<1%) fusion with *EWSR1* [113]. Approximately 60–70% of patients with the localized disease have 3–5 years overall survival, while for those with metastasis or recurrence disease, the survival remains dismal due to limited therapeutic options [113]. Aside from clinical symptomatology and radiographic imaging, no suitable tumor markers or biomarkers exist to identify and monitor patients with a high risk of recurrence, resistance to standard treatment, or early disease progression [113]. Unique pathognomonic fusions or their related transcriptome may be the detection targets that are most likely to be detected.

Ewing’s sarcoma-derived EVs were first investigated by Miller et al. [83] using RT-qPCR to detect ES-specific transcripts such as *EWSR1-FLI1*. This discovery opened the possibility of exploring the role of EV-based liquid biopsy in Ewing’s sarcoma diagnosis and monitoring of the MRD. Then, Tsugita et al. [84] discovered that EWS/FLI1 fusion mRNA, resulting from t(11;22)(q24;q12) translocation, could be identified in LEVs secreted from Ewing sarcoma cells using RT-qPCR. These fusion mRNA-LEVs were also detectable in plasma from xenografted mice [84]. This finding suggested that the EWS/FLI1 mRNA containing LEVs could be further developed as a non-invasive liquid biopsy for precise diagnosis of Ewing’s sarcoma.

### 3.7. Rhabdomyosarcoma

Rhabdomyosarcoma is the most common soft tissue sarcoma, accounting for 5–10% of sarcomas in children, adolescents, and young adults [114]. Rhabdomyosarcoma can be divided based on histology into two main types: alveolar (25%) and embryonal (75%) [114]. Alveolar rhabdomyosarcoma is characterized by chromosomal translocation t(2;13)(q35;q14), which results in a fusion between the PAX3 gene on chromosome 2 and the FOXO1 gene on chromosome 13, or in some cases, t(1;13)(p36;q14), which leads to fusion between the PAX7 gene on chromosome 1 and the FOXO gene [114]. Embryo rhabdomyosarcoma comprises cells resembling immature skeletal muscle and tends to have a more favorable prognosis than alveolar rhabdomyosarcoma [114]. The genetic abnormalities associated with embryonal rhabdomyosarcoma involve muscle development and differentiation [114].

Gayad et al. [91] conducted miRNA profiling to compare EV-derived miRNA from embryonal vs. alveolar rhabdosarcoma cell lines. The findings suggested that miRNAs in alveolar rhabdosarcoma-derived EVs expressed a distinct set of 62 miRNAs, while embryonal rhabdomyosarcoma-derived EVs were enriched with another set of 34 miRNAs [91]. Only two miRNAs (miR-1246 and 1268) were universally presented in EVs of all cell lines examined [91]. Rammal et al. [93] performed proteomic analysis between alveolar rhabdomyosarcoma-derived EVs and those of embryonal type. As a result, 81 proteins were commonly presented in EVs from both subtypes, which involved cell signaling, cell movement, and cancer progression through the integrin signaling pathway, chemokine and cytokine signaling pathway, and angiogenesis [93]. Another study reported that CD147 was exclusively expressed in metastatic tumors of human rhabdomyosarcoma tissue, which contributed to tumor cell aggressiveness, and was involved in modulating the microenvironment through rhabdomyosarcoma-derived EVs [92]. Targeted inhibition of CD147 reduced its expression in isolated EVs and suppressed their capability to enhance rhabdomyosarcoma cell invasive properties [92].

### 3.8. Pediatric Lymphoma

Lymphoma is a hematologic malignancy affecting the lymphatic system, which occurs in approximately 15% of childhood cancer [115]. According to the WHO system, lymphoma is classified as Hodgkin (HL) and non-Hodgkin lymphoma (NHL) [115]. HL is characterized by Reed–Sternberg cells and specific abnormal B lymphocytes and is primarily found in early and late adulthood but rarely in children younger than five years of age [115]. NHL originates from either abnormal B or T cells, accounts for approximately 5% of childhood cancers, and is more likely to present in younger children than HL [115]. NHL can be classified into various subtypes based on genetic markers, including Burkitt lymphoma, diffuse large B-cell lymphoma, primary mediastinal large B-cell, anaplastic large-cell lymphoma, and lymphoblastic lymphoma [116]. Pediatric lymphoma treatment typically includes the combination of chemotherapy and radiation therapy and, in some cases, may require stem cell transplantation. Early diagnosis and appropriate treatment are critical for successfully managing pediatric lymphomas.

Damanti et al. [94] investigated the role of EV-miRNAs as disease biomarkers and their functions in pediatric anaplastic large cell lymphoma (ALCL) compared to healthy donors [94]. They found miR-122-5p, a critical player in tumor cell dissemination and aggressiveness, was elevated in plasma EVs derived from ALCL patients. Of note, miR-122-5p may be a therapeutic target for ALCL pediatric patients [94]. Likewise, small RNA-sequencing analysis in plasma EVs isolated from 20 children with pediatric ALCL showed that miR-146a-5p might be useful for prognosis prediction in high-risk patients [95] since miR-146a-5p plays a role in promoting macrophage infiltration and M2-like polarization, thereby promoting ALCL aggressiveness and dissemination [95].

Lovisa et al. [97] performed a proteomics study to compare plasma EV derived from pediatric ALCL patients and healthy donors and identified up to 50 proteins that were significant in EVs derived from ALCL patients [97]. Functional enrichment analysis revealed that these proteins are related to cell adhesion, glycosaminoglycan metabolic process, extracellular matrix organization, collagen fibril organization, and acute phase response, as well as the PI3K/AKT pathway [97]. This study suggested that these ALCL-derived EV proteins can be used as diagnostic and possibly prognostic parameters during diagnosis and ALCL disease monitoring [97]. Another proteomic study identified 11 unique proteins, including five upregulated in non-relapsed HL (e.g., isoform 2 preproprotein of complement C4-A, complement C4-B, fibrinogen γ chain, inter-α-trypsin inhibitor heavy chain H2, and immunoglobulin heavy chain constant region mu) and six upregulated in relapsed HL (e.g., apolipoprotein A-I, apolipoprotein A-IV, clusterin, haptoglobin, α-1-acid glycoprotein 1, and transthyretin) [98]. Taken together, this unique set of EV proteins might be used as the biomarker to discriminate the non-relapse and relapse condition of HL patients [98].

## 4. Future Prospects

A growing number of research studies have demonstrated that EVs relate to tumorigenesis and mediate intercellular communication, which eventually leads to progression and metastasis (as summarized in Figure 2 and Table 3). EVs have been implicated in the pathogenesis and dissemination of several malignancies, as they facilitate the reciprocal communication between malignant cells and their surrounding microenvironment, hence contributing to both the advancement of the illness and the formation of metastases [117]. From this point of view, EV-based liquid biopsy with deep molecular characterization may offer mechanistic insights into tumorigenesis and progression, which will facilitate the development of personalized and precise treatment strategies.

For instance, a preclinical study unraveled a new mechanism of dinutuximab resistance in high-risk neuroblastoma: sEV-mediated inhibition of NK cell infiltration and tumor-associated macrophage (TAM) recruitment [118]. Neuroblastoma-derived sEVs suppressed splenic NK cell maturation in vivo and inhibited antibody-dependent cellular cytotoxicity mediated by dinutuximab-induced NK cells in vitro [118]. Here, RNA-sequencing of neuroblastoma sEVs identified molecular alterations involving immune effector cells, i.e., myeloid, dendritic, and NK cells, and predicted a negative regulation of leukocyte activation, thereby contributing a groundwork for neuroblastoma EV liquid biopsy to detect the immunotherapeutic resistant phenotype. Chromosomal 17q21-ter is commonly gained in neuroblastoma, where the insulin-like growth factor-2 mRNA-binding protein 1 (*IGF2BP1*) gene is located (17q21.32) [129]. *IGF2BP1* upregulation in neuroblastoma was associated with lower overall survival and positively correlated with *MYCN* mRNA, even in patients with *MYCN*-non-amplified tumors [129]. In vivo, Dhamdhere et al. [126] demonstrated the neuroblastoma EV-mediated pro-metastatic effect of *IGF2BP1*, whereas EVs from *IGF2BP1* knockdown NB cells did not trigger the metastatic process. Proteomic analysis revealed that two *IGF2BP1*-regulated proteins, SHMT2 and SEMA3A, were increased in metastatic neuroblastoma EVs. A validation study revealed the increased plasma EV levels of SHMT2 and SEMA3A proteins in patient-derived xenograft-bearing mice [126], supporting the clinical significance of IGF2BP1/SEMA3A-SHMT2 axis in neuroblastoma metastasis, which warrants further clinical studies of neuroblastoma liquid biopsy to inform therapeutic regimens against potential metastasis. Mancarella et al. [127] reported that Ewing’s sarcoma secreted EVs loaded with insulin-like growth factor 2 mRNA binding protein 3 (IGF2BP3), which promoted migration but not the proliferation of the recipient cells [127]. EV-miRNA profiling revealed that IGF2BP3 silencing of Ewing’s sarcoma cells altered specific miRNAs associated with the PI3K/Akt pathway in recipient cells [127], including miR-223-3p which target *IGF1R*, a major driver of EWS aggressiveness and a previously reported target of IGF2BP3. This finding indicated that IGF2BP3 containing EVs may have a role in regulating phenotypic heterogeneity and promoting cell migration of Ewing’s sarcoma [127].

For osteosarcoma, Macklin et al. [128] demonstrated that EVs released by cells originating from a clonal variant of the osteosarcoma cell line with high metastatic potential can be taken up by a clonal variant of the same cell line with low metastatic ability [128]. This uptake of EVs leads to the induction of a migratory and invasive phenotype [128]. The observed transfer of phenotypic traits in a horizontal manner is unidirectional, suggesting that the ability to metastasize may be developed through interclonal communication [128]. Here, proteomic analysis of EVs released by osteosarcoma clones with high metastatic potential identified several proteins relating to G-protein coupled receptor signaling pathways [128]. These findings support further investigation of EV liquid biopsy to guide precise treatment in osteosarcoma patients.

The natural ability of EVs to transport diverse bioactive compounds demonstrates their potential utility as a drug delivery system. One remarkable advantage is using EVs as biocompatible carriers for gene therapy and RNA therapeutics. Genetic engineering techniques may facilitate the encapsulation of tumor suppressor proteins into EVs. Investigating these approaches may significantly advance therapeutic strategies for pediatric cancers.

Few studies explored EV liquid biopsy and translated their findings to develop therapeutic approaches for pediatric cancers. For instance, Xue et al. [130] discovered miR-101-3p and miR-423-5p by comparing plasma EVs derived from medulloblastoma patients and healthy individuals. They found that miR-101-3p and miR-423-5p function as tumor suppressors through the direct regulation of a common gene, FOXP4, which is responsible for encoding a transcription factor that plays a crucial role in embryonic development and tumorigenesis [130]. Additionally, miR-101-3p exhibited targeting capabilities towards EZH2 (histone methyltransferase), enhancing its suppressive impact on tumor growth [130]. The use of miR-101-3p and miR-423-5p mimics significantly suppressed the proliferation, colony-forming ability, migratory ability, and invasive capability of medulloblastoma cells [130]. In addition, a xenograft nude mouse model of medulloblastoma was used to investigate the effects of overexpression miR-101-3p and miR-423-5p on tumorigenesis, which suppressed the progression of tumors [130]. In addition, Huang et al. found an elevated level of miR-130b-3p in plasma EVs obtained from medulloblastoma patients compared to healthy controls [131]. Interestingly, the miR-130b-3p mimic inhibited medulloblastoma cell proliferation by targeting serine/threonine-protein kinase 1 (SIK1), promoting the activity of the p53 signaling pathway, thus serving as a tumor suppressor miRNA in medulloblastoma [131]. These findings highlight a possibility that EV liquid biopsy may capture self-defense responses against tumorigenesis, thereby opening a new venue to investigate novel miRNA therapeutic approaches for pediatric cancers.

It should be emphasized that EVs have several advantages compared to other liquid biopsy platforms for pediatric cancer. For instance, EVs contain a variety of biomolecules, reflect tumor heterogeneity, preserve stability, and reflect the real-time change in intracellular signaling pathways that lead to identifying therapeutic targets. It is anticipated that EV liquid biopsy will make a significant contribution as a diagnostic platform for pediatric solid tumors. Nonetheless, prospective studies and validations of specific markers are required for clinical applications. To enable clinical integration of non-invasive EV liquid biopsies, especially for brain tumors where EV liquid biopsies offer transformative potential, it is essential to benchmark their diagnostic performance against established gold standard methods.

## 5. Conclusions

This review summarizes the recent progress of EV liquid biopsies in pediatric cancers and provides perspectives to explore EV biological roles, which can lead to the development of novel therapeutic approaches. The cumulative evidence suggests that EV cargo is less susceptible to degradation due to the protective role of lipid membranes surrounding vesicles. As a result, EV cargo is more stable than ctDNA and soluble proteins in bodily fluids [51,132,133]. Due to their abundance and ability to protect cargo, EVs have demonstrated useful detection sensitivities and have often been found to be superior to techniques employing ctDNA or soluble proteins as tumor antigens [51,132,133]. EV-based liquid biopsies have demonstrated promising outcomes in clinical trials, specifically for the early stage detection of cancer and the monitoring of minimal-residual disease in post-therapy follow-up patients [101,133]. Furthermore, EVs offer a similar benefit to CTCs in that they can serve as a versatile platform for detecting and measuring various types of cancer-related biomolecules. CTCs are limited by their small level in the blood, while EVs are plentiful in all human bodily fluids and offer a wide range of diagnostic possibilities [28,133]. Therefore, the analysis of EV cargo has become a promising tool for prognosis/diagnosis and is now attracting significant interest. However, there is no “gold standard” for EV isolation or characterization protocols. EV nomenclature remains inconclusive. Additionally, the challenges also affect pre-analytical procedures, including storage duration, number of freeze-and-thaw cycles and factors during transportation, which should be considered as sources of technical variations. In addition, the majority of clinical investigations of EV studies have been conducted with a relatively small cohort of patients, which lack the statistical strength of studies and the accuracy of corresponding findings [133]. In order to address the challenge mentioned earlier, the International Society for Extracellular Vesicles (ISEV) has developed the MISEV guidelines. These guidelines aim to establish a common framework for conducting studies on EVs using different protocols [57,58]. By accomplishing this, the guidelines can help minimize variations in EV detection and enhance their potential for clinical diagnosis. Prospective validation studies are needed to ensure the accuracy and reliability of EV liquid biopsies for precise diagnosis, monitoring disease progression, and guiding therapeutic intervention, which ultimately improves patient outcomes.

## Figures and Tables

**Figure 1 cancers-16-01681-f001:**
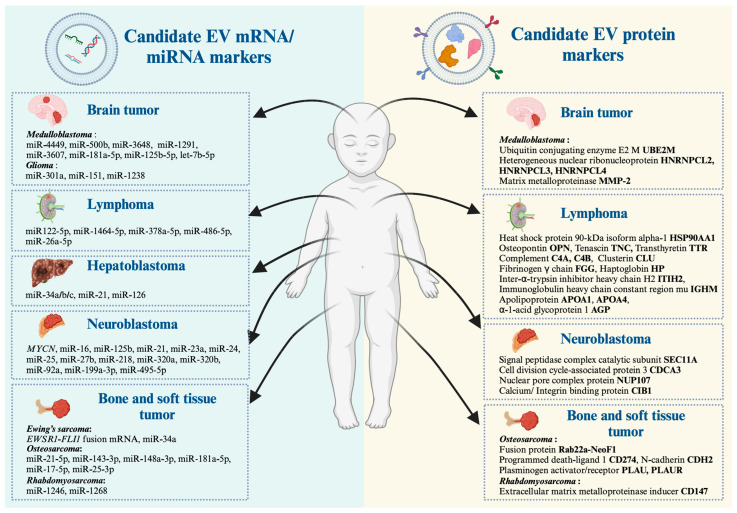
Previously identified EV biomarkers for developing pediatric cancer liquid biopsies. An illustration depicting biomolecule-containing EVs from transcriptomics and proteomics investigations in medulloblastoma (mainly arising in the cerebellum), glioma (commonly arising in the cerebral cortex), lymphoma (arising in lymph nodes of the neck, chest (mediastinum), axilla, abdomen, and groin), hepatoblastoma (in the liver), neuroblastoma (mainly arising from neural crest-derived cells), Ewing’s sarcoma (commonly arising in bones of the pelvis, legs, arms, and chest wall), osteosarcoma (mainly found in the femur, tibia, and humerus), and rhabdomyosarcoma (mainly found in skeletal muscles in the arms, legs, head, neck, and abdomen). The figure was created with BioRender.com.

**Figure 2 cancers-16-01681-f002:**
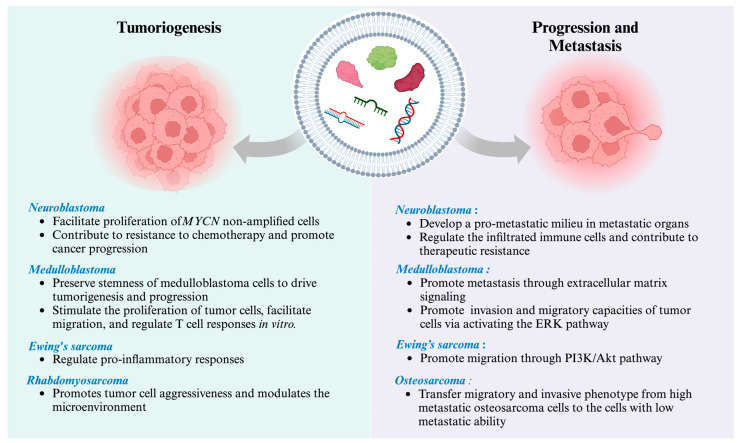
Potential roles of EVs in pediatric cancer tumorigenesis and metastasis. An illustration depicting the involvement of pathologic EVs that promote proliferation and metastasis, modulate the tumor microenvironment, and induce resistance to chemotherapy in various types of pediatric cancers. This figure was created with BioRender.com.

**Table 2 cancers-16-01681-t002:** Transcript and protein detections from EV studies of cell lines and patient-derived specimens that have been proposed as potential molecules for EV-based liquid biopsy in pediatric cancers.

Cancer Type	Vesicle Type	Transcript	Protein	Main Finding	References
Neuroblastoma	LEVs derived from *MYCN*-amplified neuroblastoma cells and patient-derived bone marrow plasma	X		*MYCN* mRNA was detectable in LEVs, but not sEVs, of *MYCN*-amplified neuroblastoma cell lines and patient-derived bone marrow plasma, where the presence and absence of *MYCN*-LEVs were associated with *MYCN* amplification status and treatment–relapse disease states.	[46]
	sEVs derived from MYCN-amplified neuroblastoma cell lines	X		miR-92a was the highest expressed in EV derived from the neuroblastoma cell line.	[70]
	sEVs derived from plasma of healthy and neuroblastoma patients	X		RT-qPCR validation discovered that sEVs (exosome) containing miR-199a-3p were significantly higher expressed in neuroblastoma patients compared with the healthy donors.	[71]
	sEVs isolated from neuroblastoma cells derived from abdominal primary tumors and bone marrow metastasis		X	Six proteins uniquely presented in metastatic neuroblastoma sEVs, e.g., signal peptidase complex catalytic subunit SEC11, cell division cycle-associated protein 3, nuclear pore complex protein Nup107, calcium, and integrin-binding protein 1. EV proteins of primary tumors are involved in neuronal development and function, while proteins exclusively present in EVs derived from neuroblastoma–bone marrow metastatic models are associated with cell survival, proliferation, and progression.	[72]
Medulloblastoma	LEVs derived from highly aggressive stem-like medulloblastoma cells overexpressing the pluripotent factor OCT4A	X	X	The interactome analysis of distinct proteins and miRNA suggested that ERK, PI3K/AKT/mTOR, EGF/EFGR, and stem cell self-renewal are the main oncogenic signaling pathways altered in these aggressive medulloblastoma cells. LEVs carried four proteins (UBE2M, HNRNPCL2, HNRNPCL3, and HNRNPCL4) and five miRNAs (miR-4449, miR-500b, miR-3648, miR-1291, and miR-3607).	[73]
	sEVs derived from group 3 medulloblastoma cell lines	X		sEVs (Exosome) derived from the group 3 medulloblastoma cell line with increased levels of miR-181a-5p, miR-125b-5p, and let-7b-5p could promote in vitro invasion and migratory abilities of a less invasive medulloblastoma cell line through the activation of ERK in the Ras/MAPK pathway.	[74]
	sEVs derived from metastatic medulloblastoma cell lines compared with non-metastatic medulloblastoma cell lines		X	The sEVs derived from the metastatic medulloblastoma cell line had significantly increased in MMP-2 localized on their external surface. This study found a high level of MMP-2 activity in CSF from three of four patients associated with tumor progression.	[75]
Glioma	sEVs derived from CSF and serum of glioma patients	X		A combination of biofluid EV-derived RNA and BEAMing RT-PCR could detect and quantify mutant and wild-type IDH1 RNA transcripts in CSF of patients with gliomas.	[76]
	sEVs derived from serum or conditioned media of glioma patients	X		Hypoxic GBM cells secrete sEVs containing miR-301a, which can promote radiation resistance in normoxia-cultured cells. Hypoxic sEVs containing miR-301a directly targeted GBM tumor suppressor TCEAL7 genes and actively suppressed their expression in normoxic glioma cells.	[77]
	sEVs derived from temozolomide (TMZ)-resistant glioblastoma multiforme cells	X		The lower expression of miR-151 in sEVs was related to an increased resistance to temozolomide (TMZ), in which the restoration of miR-151a expression sensitized TMZ-resistant glioblastoma multiforme cells.	[78]
	sEVs derived from TMZ-resistant glioblastoma cells	X		The high expression of sEVs containing miR-1238 led to the acquired resistance against temozolomide in glioblastoma-sensitive cells.	[79]
Hepatoblastoma	sEVs derived from the serum of hepatoblastoma patients	X		The level of sEV-derived miR-34a/b/c was significantly lower in the serum of patients with hepatoblastoma compared to healthy control groups.	[80]
	sEVs derived from plasma of pediatric hepatoblastoma patients	X		The elevated level of miR-21 containing sEVs in hepatoblastoma patients might be another biomarker for hepatoblastoma.	[81]
	sEVs derived from the hepatoblastoma cell line	X		sEV-derived miR-126 was upregulated in hepatoblastoma cells, which suggested that this microRNA promoted the tumorigenesis of liver cancer.	[82]
Ewing’s sarcoma	sEVs derived from Ewing’s sarcoma cell line	X		RT-qPCR detected ES-specific transcripts such as *EWSR1*-*FLI1* from Ewing’s sarcoma-derived EVs.	[83]
	LEVs derived from Ewing’s sarcoma cell of xenografted mice	X		*EWS/FLI1* fusion mRNA (resulting from the t(11;22) (q24;q12) translocation) could be identified in MVs derived from Ewing sarcoma cells and was also detectable in MVs from plasma samples of ES cell-xenografted animals.	[84]
	sEVs isolated from silenced CD99 expression of patients derived Ewing’s sarcoma cell line	X		Ewing sarcoma cells with silenced CD99 expression released EVs containing high levels of miR-34a.	[85]
Osteosarcoma	sEVs derived from malignant human osteosarcoma cell lines	X		Small extracellular vesicles derived from malignant human osteosarcomas contain miR-146a-5p.	[86]
	sEVs derived from six different human osteosarcoma or osteoblast cell lines	X		Next-generation miRNA sequencing revealed miRNAs in cell lines with different degrees of metastatic potential and found that mi-21-5p, miR-143-3p, miR-148a-3p, and 181a-5p are highly expressed in sEV-derived metastatic SAOS2 cells.	[87]
	sEVs derived from the osteosarcoma cell line	X		Circulating miR-17-5p and miR-25-3p could be identified in osteosarcoma cells, which are used as a novel diagnostic and prognostic biomarker and also reflect tumor burden in the osteosarcoma mouse model.	[88]
	sEVs derived from the Rab22a-NeoF1 fusion protein cell line		X	The osteosarcoma Rab22a-NeoF1 fusion protein was secreted via EV by binding to the KFERQ-like motif of HSP90, which was taken up by macrophages and other cancer cells. EV derived from osteosarcoma contained programmed death-ligand 1 (PD-L1) and N-cadherin.	[89]
	sEVs derived from the metastatic osteosarcoma cell line		X	The elevation of uPA and its receptor (uPAR) was cargo via sEVs secreted from metastatic osteosarcoma cells.	[90]
Rhabdomyosarcoma	sEVs derived from rhabdomyosarcoma cell lines	X		Ten miRNAs were common among the two rhabdomyosarcoma cell lines (JR1 and RD), while only two miRNAs (miR-1246 and miR-1268) were present in EVs of all cell lines.	[91]
	sEV derived from the rhabdomyosarcoma cell lines		X	CD147 was exclusively expressed in metastatic tumors of human rhabdomyosarcoma tissue, which was involved in modulating the microenvironment through rhabdomyosarcoma-secreted sEVs.	[92]
	sEV derived from alveolar and embryonal rhabdomyosarcoma cell lines		X	A proteomic study revealed 122 common proteins in alveolar rhabdomyosarcoma-derived EVs and 161 common proteins in embryonal rhabdomyosarcoma-derived EVs. The biological process analysis suggested that 81 proteins were common to both subtypes, which involved cell signaling, cell movement, and cancer.	[93]
Pediatric lymphoma	sEVs derived from plasma of ALCL patients	X		miR-122-5p was elevated in plasma sEVs derived from ALCL patients, which are critical in promoting tumor cell dissemination and aggressiveness.	[94]
	sEVs derived from plasma of 20 pediatric anaplastic lymphoma kinase-positive ALCL patients	X		Small RNA-sequencing analysis in plasma sEVs from 20 pediatric ALCA suggested that miR-146a-5p and miR-378a-3p showed a negative prognostic impact in both univariate and multivariate analysis.	[95]
	sEVs derived from Burkitt lymphoma, Hodgkin lymphoma, and mature B-cell acute lymphoblastic leukemia	X		Burkitt lymphoma, Hodgkin lymphoma, and mature B-cell acute lymphoblastic leukemia most stably expressed miR-26a-5p and miR-486-5p in sEVs.	[96]
	sEVs derived from plasma pediatric ALCL patients and healthy donors		X	The Reactome database and KEGG networks highlighted a dramatic increase in proteins of the PI3K/AKT pathway in ALCL-sEVs, which included heat shock protein 90-kDa isoform alpha-1, osteopontin, and tenascin plasma EV derived from pediatric ALCL patients.	[97]
	sEVs derived from plasma of non-relapsed and relapsed nodular sclerosis Hodgkin lymphoma		X	LC-MS/MS identified these 11 unique protein spots, including five more abundant in non-relapsed HL (e.g., isoform 2 preproprotein of complement C4-A, complement C4-B, fibrinogen γ chain, inter-α-trypsin inhibitor heavy chain H2, and immunoglobulin heavy chain constant region mu) and six more abundant in relapsed HL (e.g., apolipoprotein A-I, apolipoprotein A-IV, clusterin, haptoglobin, α-1-acid glycoprotein 1, and transthyretin).	[98]

**Table 3 cancers-16-01681-t003:** The relationship between EV-based liquid biopsy, their molecular cargos, and functional involvement in childhood cancer tumorigenesis and metastasis.

Disease Stage	Disease	Significant Biology Process	Vesicle Type	EV Molecular Cargo	Main Finding	References
Tumorigenesis	Neuroblastoma	Regulation of infiltrated immune cells after chemotherapy treatment	sEVs	Not specified	Small EVs suppressed splenic NK cell maturation in vivo and dinutuximab-induced NK cell-mediated antibody-dependent cellular cytotoxicity in vitro upon dinutuximab treatment.	[118]
		Facilitating the migration and proliferation of non-*MYCN* amplified cells	sEVs released by *MYCN*-amplified neuroblastoma cells	miR-17-5p	miR-17-5p is crucial in facilitating the migration and proliferation of non-*MYCN* amplified cells.	[119]
		Contribute to resistance to chemotherapy and promote cancer progression	sEVs derived from neuroblastoma cell line	miR-21, miR-155	The data presented in this study demonstrate the distinct function of EV-derived miR-21 and miR-155 in intercellular communication between neuroblastoma cells and human monocytes, contributing to developing resistance to chemotherapy and cancer progression.	[120]
	Ewing’s sarcoma	Regulation of pro-inflammatory response	sEVs derived from Ewing’s sarcoma cell lines	Not specified	Exposure to Ewing’s sarcoma EVs inhibited the process of cellular development towards moDCs, as indicated by the decreased expression levels of co-stimulatory molecules (e.g., CD80, CD86, and HLA-DR). Ewing’s sarcoma EVs exhibited the ability to suppress the proliferation of CD4^+^ and CD8^+^ T cells, as well as the release of IFNγ, while simultaneously increasing the secretion of IL-10 and IL-6.	[121]
	Rhabdosarcoma	Promotes tumor cell aggressiveness and modulates the microenvironment	sEV derived from rhabdomyosarcoma cell lines	CD147	Treatment of normal fibroblasts with rhabdomyosarcoma-derived EVs increased proliferation, migration, and invasion, whereas CD147-downregulated rhabdomyosarcoma cells block these effects.	[92]
	Medulloblastoma	Preserve stem cell characteristics or communicate with neighboring cells to promote the progression of group 4 of medulloblastoma	sEVs derived from bulk tumor cells and brain tumor sheroid-forming cells	miR-135b, miR-135a	The suppression of miR-135b and miR-135a leads to a decrease in the stemness of medulloblastoma brain tumor spheroid-forming cells, in which AMOTL2 was targeted by miR-135b and miR-135a.	[122]
		Promote in vitro invasion and migratory capacities of tumor cells via activating the ERK pathway in the Ras/MAPK signaling cascade	sEVs derived from Group 3 medulloblastoma cell lines	miR-181a-5p, miR-125b-5p, let-7b-5p	The upregulation of these miRNAs led to more significant in vitro invasion and migratory capacities of tumor cells via activating the ERK pathway in the Ras/MAPK signaling cascade.	[74]
		Stimulate the proliferation of tumor cells, facilitate migration, and regulate T cell responses in vitro	sEVs derived from medulloblastoma cell lines	Not specified	The examination of the functional properties of EVs can stimulate the proliferation of tumor cells, facilitate migration, and regulate T cell responses, which might play a crucial role in the progression of medulloblastoma.	[123]
		Play a role in the progression of medulloblastoma	sEVs derived from control and overexpressed B7-H3 cells	B7-H3 (immunosuppressive immune check point)	This study revealed a novel role in EV production and packaging for B7-H3 that may contribute to medulloblastoma progression.	[124]
		Involvement in the advancement and invasion of medulloblastoma.	sEVs derived from medulloblastoma cell line	Iron carrier proteins	This study establishes the relationship between iron metabolism and the advancement and invasion of medulloblastoma. The reduction in iron induces cell cycle arrest in the G1/S phases, resulting in the suppression of cell proliferation and the initiation of apoptosis.	[125]
Cancer EVs promote metastasis	Neuroblastoma	Pro-metastatic	sEVs derived from neuroblastoma cell line	IGF2BP1	IGF2BP1 affects the levels of SEMA3A and SHMT2 in EVs, and this regulation plays a role in developing a pro-metastatic milieu in metastatic organs.	[126]
	Ewing’s sarcoma	Promote tumor cell migration	sEVs derived from Ewing’s sarcoma cell line	IGF2BP3	This data indicated that IGF2BP3 containing EVs may have a role in regulating phenotypic heterogeneity and promoting cell migration of Ewing’s sarcoma.	[127]
	Osteosarcoma	Promote tumor cell migration and invasive phenotype	sEVs derived from high metastatic potential cell line	Several proteins relating to signaling pathways mediated by G-protein coupled receptors	The uptake of sEVs derived from the high metastatic osteosarcoma cell line by the same cell line with low metastatic ability leads to the induction of migratory and invasive phenotypes. Proteins relating to signaling pathways mediated by G-protein coupled receptors may play a crucial role in driving metastasis of osteosarcoma.	[128]
	Medulloblastoma	Promote medulloblastoma metastasis through extracellular matrix signaling	sEVs derived from metastatic medulloblastoma cell line	EMMPRIN, MMP-2	This study provides evidence for the significance of EMMPRIN and MMP-2-associated sEVs in facilitating a conducive environment that promotes medulloblastoma metastasis.	[75]

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
