# Peer review of "Extracellular Vesicles for Childhood Cancer Liquid Biopsy"

_cancers, 2024, doi:10.3390/cancers16091681_

Round 1

Reviewer 1 Report

Comments and Suggestions for Authors

In their review article, Singhto and colleagues outline the potential applications of using extracellular vesicles to detect and monitor various childhood cancers. The review is informative and aligns with the journal's scope.

Minor issues should be addressed:

1.     Although medulloblastomas are primary brain tumors, it should be mentioned they arise in or related to the cerebellum. Also, the figure 1 (top right) is slightly misleading in this respect, not depicting the cerebellum, but mentioning only medulloblastoma.

2.     The authors could have included one of the highly cited reviews from Cancers on liquid biopsy and primary brain tumors, including medulloblastomas.

3.     Line 270-278 Appears slightly wrong reference #89 in some cases: consider, perhaps, citing ref. 1 from the cited reference, i.e.: “Louis DN et al. 2021”, or the real “WHO classification of tumors of the nervous system” (2021), when it comes to the four molecularly defined groups of medulloblastoma.

4.     As an example, it could have been mentioned that the detection of p53 mutations may be monitored in medulloblastoma, where SHH (group 2) is divided by the WHO for p53-mutant, and wildtype, which has relevance for prognosis.

5.     The title of Table 2 may be misleading since most cited findings relate to cell lines rather than patients. A suggestion is to clarify that the table includes findings from cell lines, (which then isn’t liquid biopsy).

6.     Line 473: are they really “essential”, or just “supporting”, or “related” to tumorigenesis?

7.    There should be a paragraph included on the limitations of EVs, like how far away their clinical use is, or how little the exact molecular diagnosis may help without real treatment options for some tumors.

Reviewer 2 Report

Comments and Suggestions for Authors

Extracellular vesicles for childhood cancer liquid biopsy

Minor revisions: 

1. The introductory section could benefit from including more references, as some passages describing various research results seem to have only a single citation. Adding more papers would increase the value of this review. 

2. Please standardize references throughout the text according to a chosen format. 

3. For all figures, it would be beneficial to expand the information box to include more information and description to help readers understand the visual data presented.

4. As there is no Discussion section, the Conclusion seems a bit short for a large review paper. This section is intended to help organize thoughts and guide the reader, it is intended to highlight the significance, importance and relevance of the work. It should focus on explaining and evaluating the information presented in the review, rather than simply providing a synopsis of the previous chapters. It should also provide an argument to support the conclusion. The conclusion provides a closure for the reader and reminds them of the content and importance of the research. Therefore, it is recommended to expand the last section to include the points mentioned above. 
